# Post-Islamism and Intellectual Production: A Bibliometric Analysis of the Evolution of Contemporary Islamic Thought

**Mohamed Amine Brahimi [1,\*]** and **Houssem Ben Lazreg [2,\*]**

1 Department of Sociology, Columbia University, New York, NY 10027, USA
2 Department of Modern Languages & Cultural Studies, University of Alberta, Edmonton, AB T6G 2E6, Canada
\* Correspondence: mab2434@columbia.edu (M.A.B.); benlazre@ualberta.ca (H.B.L.)

**Abstract:** The advent of the 1990s marked, among other things, the restructuring of the Muslim world in its relation to Islam. This new context has proved to be extremely favorable to the emergence of scholars who define themselves as reformists or modernists. They have dedicated themselves to reform in Islam based on the values of peace, human rights, and secular governance. One can find an example of this approach in the works of renowned intellectuals such as Farid Esack, Mohamed Talbi, or Mohamed Arkoun, to name a few. However, the question of Islamic reform has been debated during the 19th and 20th centuries. This article aims to comprehend the historical evolution of contemporary reformist thinkers in the scientific field. The literature surrounding these intellectuals is based primarily on content analysis. These approaches share a type of reading that focuses on the interaction and codetermination of religious interpretations rather than on the relationships and social dynamics that constitute them. Despite these contributions, it seems vital to question this contemporary thinking differently: what influence does the context of post-Islamism have on the emergence of this intellectual trend? What connections does it have with the social sciences and humanities? How did it evolve historically? In this context, the researchers will analyze co-citations in representative samples to illustrate the theoretical framework in which these intellectuals are located, and its evolution. Using selected cases, this process will help us to both underline the empowerment of contemporary Islamic thought and the formation of a real corpus of works seeking to reform Islam.

**Keywords:** reform; Islamic thought; bibliometric analysis; post-Islamism; references; scientific and intellectual field

## 1. Introduction

The advent of the 1990s marked, among other things, the restructuring of the Muslim world in its relationship to Islam. According to Olivier Roy, "the Muslim world has entered the 'post-Islamist' era, defined as the appearance of a space of secularism in Muslim societies (...) the religious field (tending) to dissociate itself from the political field" (Roy 1999, p. 13). This new context has proved extremely favorable to the emergence of new scholars who define themselves as reformists or modernists[1]. During the 2000s, there was a proliferation of various monographs and biographies on these contemporary Islamic thinkers. (Abu-Rabi' 2006; Benzine 2008; Kurzman 1998; Fillali-Ansary 2005; Hunter and Hunter 2014).

Enthusiasm for this new trend, which views itself as a reformed Islam, unveiled a growing interest in the academic field of Islamic studies. The reconfiguration of the Islamic intellectual world through renewed academic production, public discussions, and intellectual debates, brings to light the reality of a profound shift. This mainly affects the

---

1 It seems important to note that Olivier Roy questions in his article the possible correlation between the post-Islamist moment and the birth of a theological reformulation, which seems to us to be in strong opposition to the thesis that we defend in this text.



conceptualization of religion as a tool for modernizing society. These scholars approach religion in a new way: they aspire to spread the message that Islam encompasses a liberal way of thinking in line with modernity and is responsive to the reality of a multi-religious and multicultural world. The article aims to account for the historical evolution of these intellectuals in the scientific field.

However, the question of Islamic reform has been debated during the 19th and 20th centuries. Some studies refer to these contemporary Islamic thinkers as adherents to the reformist current of the late nineteenth century (Benzine 2008). It is in a colonial context, marked by the incursion of European powers into Muslim societies and the undermining of traditional religious authorities, that reformism first emerged. Colonial powers and orientalists such as Ernest Renan define Islamic civilization as a subaltern to deny the claim of Muslims to equality. In response to this discourse, the reformist movement came to the fore in almost all Muslim societies as an alternative that aims at modernizing and renewing religious conceptions (Hourani 1983).

This movement of renewal was followed by Islamism, a political trend driven by a theocentric and traditional conception of Islam. The latter was espoused by groups that thrived in the societies of postcolonial Islam (Kepel 1993; Roy 2011). It is in response to this political instrumentalization of religion and through the desire to recapture or overtake the essence of the first reformists that contemporary Islamic thought has emerged. One of the key moments of this critical reflection was the Iranian revolution in the early 1980s, which inspired the intellectual elites of the Muslim world to rethink the interpretation of Islamic scripture (the Quran) and other texts (the Hadith, Fiqh, etc.). However, scholars did not begin engaging with the Western public space until the 1990s (Zeghal 2008). In this context, the media coverage of certain events related to Islam and Muslims, such as the Salman Rushdie controversy, the Islamic veil controversy, and the 11 September attacks, played a seminal role in inducing these players to promote a new religious thought.

One of the contemporary reformist thinkers' main characteristics is that they are very active in scientific fields compared to their predecessors. Using tools from the social sciences, these scholars place the Islamic message back in its historical context and seek to develop a truly universal message in a dialogue with other religions. With contemporary knowledge—such as linguistics, philosophy, social sciences, etc.—they address issues as diverse as the hermeneutics of the Quran, the construction of renewed religious norms, or gender equality issues in order to modernize Islam.

The literature surrounding contemporary Islamic thought is based primarily on content analysis. Despite initially being developed to be applied in different disciplines, these approaches share a type of reading that focuses on the interaction and codetermination of religious interpretations rather than on the relationships and social dynamics that constitute them (Marcotte 2010). Based on these approaches, the act of reinterpretation of traditional texts (the Quran, the Hadith) is considered to be the driving force behind this current of thought (Taji-Farouki 2006; Rahemtulla 2017). These biographical writings addressing Islamic thought emphasize their link with contemporary human sciences; their subjects' contributions are often seen as tools of religious reform (Fillali-Ansary 2005; Benzine 2008). It must be noted that these analyses remain in the orbit of textuality as they do not raise the social issues implied by the production of knowledge. Despite their contribution, it seems important to question this contemporary thinking differently: what influence does the context of post-Islamism have on its emergence? What connections does it have with the social sciences and humanities? How did it evolve historically? To answer these questions, we used a methodology that is not common in the study of contemporary Islamic thought: bibliometrics. The first step is to explain our method and the way in which we selected a representative sample. Then, with the help of co-citations analysis, we discern the evolution of Muslim modernist intellectuals through their scientific productions. Our main purpose is to delineate the theoretical framework in which these intellectuals are located, as well as its evolution. Using selected cases, this process will help us to both underline



the empowerment of contemporary Islamic thought and the formation of a real corpus of works seeking to reform Islam.

## 2. Methodology

To better understand the dynamics that characterize contemporary Islamic thought, we have chosen to adopt a sociological approach. On the topic of intellectual production, sociology has undergone a considerable renewal[2]. In its attempts to describe the logic of competing stakes and the rules that promote or demote the participants of a given academic field, sociology seeks to objectify the dynamics of social recognition. Through a material approach to culture, sociology aims to explain the influences and theoretical transformations in connection with intellectual practices. In a world marked by unequal access to spaces of symbolic production, every scholar is forced to adapt their strategies according to possible alternative expectations, the nature of which varies from field to field.

It may be noted that this type of analysis, centered on the objectification of intellectual practices, is rarely applied regarding our subject. Content analyses are necessary but must be supplemented by sociological tools, the aim being to offer the most complete portrait of the Muslim intellectual world.

This paper uses a bibliometric analysis of scientific articles as well as citations from these scholars of Islam. (see Appendix A). These citations are an indicator of recognition or a lack thereof in the intellectual field. Far from any idea of pure science, publication in specialized journals, like any social practice, targets strategic interests; the goal being the acquisition of authority to maximize strictly symbolic benefits. This is how the number of citations becomes an indicator of scientific credibility within the intellectual field (Bourdieu 1975).

Bibliometrics is a set of statistical methods that have been used in books, articles, and other publications. It enables the analysis of publications and their characteristics (Gingras 2016). Bibliometrics is applied in the context of disciplines that view science and scientific communities as objects of study, such as economy of knowledge, sociology of sciences, epistemology, and the history of science. By tracking the situations in which authors refer to each other, it is possible to assess the degree of interaction, as well as the visibility, influence, and partnerships in the academic community (Pritchard 1969). The referential universe of a scientific article refers to the highly relational aspect of the intellectual field; the actors in this social space plan to exchange a specific production (articles, theories, knowledge, etc.) for recognition (citations, prices, jobs, etc.). This sense of giving and taking implied by this symbolic barter is based on the esteem of the intellectual community. The act of publishing is the foundation of these links; a researcher who publishes an article in a recognized journal contributes to the advancement of a field of research. Through publication, the community allows an article to be recognized by peers through citations (gift/counter-gift) (Hagstrom 1975).

Within this framework, the citation remains a central element in the description of affiliations within academia and making it possible to evaluate the intellectuals in relation to their scientific productions. To compile our corpus, we chose the Web of Science[3] database, which contains 9000 journals, accessible online via subscription. It offers a wide range of journals available in full, selected according to explicitly defined criteria[4]. Bibliometrics provides a realistic portrait of intellectual production which goes beyond the arbitrary rankings that can be generated by a simple content analysis. Bibliometrics provides a meaningful overview of the different trends that characterize Islamic thought and its evolution within the scientific community.

In this context, our analysis will be structured according to the following criteria:

---

2   Examples include the works of Pierre Bourdieu, Charles Camic, and Randall Collins (Camic 1983; Bourdieu 1988; Collins 1998).

3   The Web of Science is a university database.

4   Established scholars select the journals to be covered in cooperation with users, publishers, and members of editorial boards. Selection criteria include publication frequency, compliance with international presentation conventions, and a peer review committee (Archambault et al. 2013).

1.  Selection of a group of authors who are representative of contemporary Islamic thought. Although our sample is restricted to selected intellectuals who have produced scientific papers, we acknowledge that some important contemporary reformist thinkers such as Abdolkarim Soroush are not included.
2.  Analysis of the co-citations to explain the theoretical background in which the authors are embedded, and its evolution.
3.  Interpretation of the results to trace the evolution of contemporary Islamic thought in the scientific field.

## 3. Selection of Authors

Islamic thought is a challenging field of study because its scholarly community is neither clearly defined, nor is its object of study. Indeed, the one factor uniting this group of authors is the desire to reform Islam with help from the social sciences (linguistics, semiology, comparative history of religions, sociology, etc.). At first glance, the covered topics are diverse and range from the study of the Quran to the traditions or history of Islam, and thus, are not identified as being part of a specific field of study or discipline.

These intellectuals adopt an approach that calls into question the division between sanctioned theological discourse and scientific neutrality. Using scientific method tools such as hermeneutics, linguistics, and semiotics, they challenge clerics' monopoly of the interpretation of religious norms and incite the scientific enterprise of religious studies to be more committed to transforming religion. Sociologically, this type of stance engenders participation across several fields where different forms of symbolic capital are used. For example, reformers hold a symbolic capital that allows them to be reliable references at the academic level and confers them the legitimacy to be political advisors. Likewise, their theological stances make them prophetic figures in religious spaces or Islamic community associations. When deemed experts by those in power, or as independent commentators, they publicly take stances on Islam-related issues.

The study of these polymorphic actors involves implementing a "sequential" sampling procedure (Miles and Huberman 2003, p. 59), which requires continuous evaluation to reach a valid level of representation and coherence. As the sample size is not preset in advance, it is necessary to track the prevalent measuring units in the population under study and interrupt or finalize the selection when the information is satisfactory. The core corpus consists of the main works that deal with the question of contemporary Islamic reform[5]. While diverse, these writings are introductions to the current state of Islamic thought, and most of them present biographical or intellectual accounts in connection with multiple themes. A total of 41 books were identified (see Table 1), the majority of which were published starting in the 1990s in English or French.

At the end of this process, we established a list of 226 scholars mentioned in these works. The list then underwent three stages of adjustment, described below.

We made a first selection by targeting the authors who prioritize the promotion of a re-reading of Islam to modernize it; this has helped discard those who do not outline any considerations about Islamic reform. Some books show a tendency to categorize several Muslim scholars as reformers, even if they do not go beyond summarizing and introducing their religion to the lay public or otherwise engaging in community activism.

---

[5]  This search was conducted using three search engines: WorldCat, HOLLIS +, and Google Books.

**Table 1.** 40 Books Collected to Select our Study Population.

Abu-Rabi', Ibrahim. The Blackwell Companion to Contemporary Islamic Thought. 1st edition. Malden, Mass: Wiley-Blackwell, 2006.

Abu-Rabi', Ibrahim M. Contemporary Arab Thought: Studies in Post-1967 Arab Intellectual History. Pluto Press, 2004.

Ahmed, Safdar. Reform and Modernity in Islam: The Philosophical, Cultural and Political Discourses Among Muslim Reformers. I.B.Tauris, 2013.

Akhter, Shamim. Faith & Philosophy of Islam. Gyan Publishing House, 2009.

Aksikas, Jaafar. Arab Modernities: Islamism, Nationalism, and Liberalism in the Post-Colonial Arab World. Peter Lang, 2009.

Amirpur, Katajun. New Thinking in Islam: The Jihad for Democracy, Freedom and Women's Rights. Gingko Library, 2016.

Bayat, Asef. Making Islam Democratic: Social Movements and the Post-Islamist Turn. Stanford University Press, 2007.

Bennett, Clinton. Muslims and Modernity: Current Debates. A&C Black, 2005.

Benzine, Rachid. Les nouveaux penseurs de l'Islam. Paris: Editions Albin Michel, 2008.

Browers, Michaelle, et Charles Kurzman. An Islamic Reformation? Lexington Books, 2004.

Corm, Georges. Pensée et politique dans le monde arabe. Paris: La Découverte, 2015.

Esposito, John L. The Future of Islam. Oxford University Press, 2010.

Esposito, John L., et John Voll. Makers of Contemporary Islam. 1st edition. New York: Oxford University Press, 2001.

Filali-Ansary, Abdou. Réformer l'islam? Une introduction aux débats contemporains. Paris: Editions La Découverte, 2005.

Hafez, Kai. Radicalism and Political Reform in the Islamic and Western Worlds. Cambridge University Press, 2010.

Hafez, Ziad. La pensée religieuse en islam contemporain: Débats et critiques. Paris: Librairie orientaliste Paul Geuthner, 2012.

Hidayatullah, Aysha A. Feminist Edges of the Qur'an. Oxford University Press, 2014.

Hunter, Shireen, et Shireen T. Hunter. Reformist Voices of Islam: Mediating Islam and Modernity. Armonk, N.Y.: Routledge, 2014.

Jackson, Roy. Fifty Key Figures in Islam. Routledge, 2006.

Kamrava, Mehran. The New Voices of Islam: Rethinking Politics and Modernity: A Reader. University of California Press, 2006.

Kassab, Elizabeth Suzanne. Contemporary Arab Thought: Cultural Critique in Comparative Perspective. New York: Columbia University Press, 2009.

Khoury, Paul. Tradition et modernité: Thèmes et tendances de la pensée arabe contemporaine (les années 60 et 70)-Troisième édition. Editions L'Harmattan, 2013.

Kurzman, Charles. Liberal Islam: A Source Book. Oxford University Press, 1998.

Lahoud, Nelly. Political Thought in Islam: A Study in Intellectual Boundaries. Routledge, 2013.

Laurence, Jonathan, et Justin Vaïsse. Integrating Islam: Political and Religious Challenges in Contemporary France. Brookings Institution Press, 2007.

Leaman, Oliver. History of Islamic Philosophy. Routledge, 2013.

Lovat, Terence. Women in Islam: Reflections on Historical and Contemporary Research. Springer Science & Business Media, 2012.

Macqueen, Benjamin. Islam and the Question of Reform: Critical Voices from Muslim Communities. Academic Monographs, 2008.

Marcotte, Roxanne D. Un islam, des islams? Editions L'Harmattan, 2010.

Mervin, Sabrina. Les mondes chiites et l'Iran. KARTHALA Editions, 2007.

Mir-Hosseini, Ziba, et Richard Tapper. Islam and Democracy in Iran: Eshkevari and the Quest for Reform. I.B.Tauris, 2006.

Moaddel, Mansoor, et Kamran Talattof. Modernist and Fundamentalist Debates in Islam: A Reader. 1st edition. New York: Palgrave Macmillan, 2002.

Ourya, Mohamed. La pensée arabe actuelle: Entre tradition et modernité. Editions L'Harmattan, 2016.

Rahemtulla, Shadaab. Qur'an of the Oppressed: Liberation Theology and Gender Justice in Islam. Oxford University Press, 2017.

Saeed, Abdullah. Reading the Qur'an in the Twenty-First Century: A Contextualist Approach. Routledge, 2013.

Safi, Omid. Progressive Muslims: On Justice, Gender, and Pluralism. Oneworld Publications, 2003.

Taji-Farouki, Suha. Modern Muslim Intellectuals and the Qur'an. OUP, 2006.

Tibi, Bassam. Islam's Predicament with Modernity: Religious Reform and Cultural Change. Routledge, 2009.

Waardenburg, Jean Jacques. Muslims as Actors: Islamic Meanings and Muslim Interpretations in the Perspective of the Study of Religions. Walter de Gruyter, 2007.

The selection that followed led to removing of religious leaders from the initial sampling, namely the Ulama[6] and the Imams[7]. These religious authority figures operate within constraints and with a different legitimacy from that conferred on an intellectual. Like Weberian priests, Ulama and Imams are mandated and certified by religious institutions (Weber 2011, pp. 50–54). Similarly, the ideologues of political Islam have also been dismissed since their discourse differs from that of the participants under study[8]. Ideologues

---

[6]  A body of Muslim scholars who are recognized as having specialist knowledge of Islamic sacred law and theology.

[7]  An Imam is a person who leads congregational prayers in a mosque. He is also a religious authority in the community he officiates.

[8]  In the case of Islamists, we have to deal with an approach where the religious heritage is reinterpreted to guide political action. See the example of Sayyid Qutb, the ideologue of the Muslim brothers (Carré 1984).

utilize the term "reform", but always to revitalize religious referents (Caliphate, Sharia, etc.) to promote political action.

We performed the final selection based on a more systematic reading of the writings of our study population. From this content analysis, we kept only scholars who have an insider's understanding of Islam, whereas we excluded those who offer a critical look at their Islamic heritage using modern sciences. The chosen intellectuals are those who are directly involved with religious tradition and whose writings are not reducible to mere critical research. They adopt a scientific approach that fully corresponds to the production of new Islamic norms. We, therefore, mobilized the resulting knowledge to address developing issues of a theological order. This type of scientific approach also gives these intellectuals legitimacy in academic spaces. For example, intellectuals like Mohamed Talbi, Khaled Abou El Fadl, or Amina Wadud play both sides. They are scholars, and at the same time, they position themselves within the religious field. They want to reshape the practice of Muslim communities. We have also selected academics who have published scientific articles; this has drastically limited our sample because this type of publication is not prioritized by all modernist intellectuals, but only by those who are scientifically active. Therefore, our study population includes 23 individuals and a total of 201 articles (see Table 2), mostly published in French and English. These authors share two characteristics in their discourse: first, they call for an internal engagement with Islam, which can be justified in different terms (authenticity, civilization autonomy, and the quest for religious specificity); second, they do it freely while transcending the constraints of any political, religious, or scientific orthodoxy.

**Table 2.** The 23 Authors Selected for Bibliometric Analysis.

| |
|---|
| Farid Esack |
| Hassan Hanafi |
| Abdelwahab Meddeb |
| Ebrahim Moosa |
| Hossein Nasr |
| Mohamed Talbi |
| Asghar Ali Engineer |
| Chandra Muzaffar |
| Abdullahi Ahmed An-Na'im |
| Fathi Osman |
| Louay M. Safi |
| Nasr Hamid Abu Zayd |
| Abdelmajid Charfi |
| Malek Chebel |
| Souleyman Bachir Diagne |
| Khaled Abou El Fadl |
| Tariq Ramadan |
| Ziauddin Sardar |
| Youssef Seddik |
| Shabbir Akhtar |
| Mohamed Arkoun |
| Abdennour Bidar |
| Amina Wadud |

## 4. The Space of Reference for Islamic Thought

A quantitative demonstration of the recognition of Islamic thinkers prompts us to scrutinize the effects of this recognition regarding their theoretical references: how they adapt to the scientific field. To examine this aspect, it is necessary to reconstruct the referential space in which these scholars evolve. This reconstruction enables us to go beyond a simple reading stimulated by personal experiences as a primary source. Beyond the simple "empirical individual" perceivable by common sense, it is important to be able

to reconstruct the homologies as well as the relations which characterize the "epistemic individuals" (Bourdieu 1984).

For this purpose, co-citations are an important instrument as they show the knowledge structure of a discipline or a group of authors. Co-citation is defined as "the frequency with which two items of earlier literature are cited together by the later literature" (Small 1973, pp. 265–66). Developed by Small and Griffith (Small and Griffith 1974), this technique assumes that there is an intellectual relationship between each pair of documents in the body of the text. The importance of a specific co-citation relationship lies in the frequency of its occurrence throughout the sample. This indicator makes it possible to reconstruct the referential space by identifying the relationships between the researchers and the research topics that appear in the corpus of citations. The authors' names are neither empirical individuals nor real social actors, but proper names that carry a theoretical meaning for the authors who mobilize them (Kreuzman 2001). In this frame, the citation is a way of expressing a position in the scientific world.

To conduct this analysis, we collected citations in the articles of the targeted authors, which yielded a list of 218 articles. Out of this entry, we created a co-citation matrix spread over three 10-year periods (1980–1990, 1991–2001, 2002–2012); this enables the measurement of the historical development in the references used. The logic behind the choice of these historical cycles is related to statistical as well as social considerations. The first issue we had to look at is the search for a similar temporality for each period. This methodological constraint forced us to divide the periods into decades. From a socio-political point of view, each period refers to events that considerably reshaped the Muslim world. They are the founding moments of particular "historical cycles[9]," where the multiple representations of Islam are deeply restructured. The period from 1980 to 1990 starts with the Iranian revolution (1979), then the 1990s (1991–2001) are marked by the Salman Rushdie affair, and finally the years 2000 (2002–2012) are defined by the attacks of 11 September.

We employed network analysis to visualize the most important links of our matrix. To ensure the most accurate reading possible, we have established a co-citation threshold for each graph, which helps us select the most relevant authors. The objective here is to elucidate the different theoretical relationships that an article contains in terms of nodes and links (Wasserman and Faust 1994). While the nodes represent the authors cited in the network, the links between these nodes show the presence of two references in the same textual corpus. The number of co-citations is illustrated by two parameters: the size of the nodes and the number of links between them according to a pre-established threshold. Out of this initial layout, an analyzable structure emerges where all the links between the nodes carry significant information.

One of the practical benefits of network analysis is that it helps to understand and visualize the linking structures for a set of variables. This graphic representation offers a general overview of the links between different references while reproducing the space of their relations. To process the bibliometric database, we used the Louvain algorithm. This methodology allows us to identify the structure of what is commonly called "large-scale graphs"[10], generally made up of sub-graphs which are highly interconnected into "communities". The purpose of the algorithm is to maximize the modularity for partitioning a graph, which implies ensuring that the number and weight of the links are larger within communities (intra-community density) than between communities (intercommunity density)[11]. The Louvain algorithm makes it possible to raise the value of intra-community links to provide a better view of the different communities within the total mass of links. It operates from the bottom up, in the sense that small structures with good modularity

---

[9] This term is borrowed from Hamit Bozerslan who developed it to better understand the different temporalities of the Middle East's history; see (Bozarslan 2011).

[10] These graphics do not have a simple apparent structure (like clusters or trees); see (Latapy 2007).

[11] Modularity is a measure for the quality of partitioning the nodes of a graph or network in communities. It also enables some community detection tasks in graphs; see: (Newman 2006).

are not likely to be embedded in larger and less significant structures. Using this type of tracking allows us to include many networks and to maximize the use of our co-citation matrix. For all the networks, we retain nodes of a certain degree. This degree represents the total number of links that the node maintains with another node. If reference A is connected exclusively to a reference B, their link corresponds to a degree of 1. The degree does not take into account the weight of the link (the number of co-citations between A and B). Nodes with the highest degrees can be identified through calligraphic shades. Thus, the names of the most co-cited authors are written with larger fonts. The selection of the most representative degree level is made automatically by the algorithm to create the most representative graph. The degree changes from one graph to another because the number of articles varies. Our main goal is to have the best visibility, so we can easily recognize the community.

### 5. First Period (1980–1990): Dominance of Orientalism[12]

For the first period, we analyzed a total of 63 articles written by authors who published their work between 1980 and 1990. The Louvain algorithm that was applied to retain the node has 22 degrees or more. Our dataset's most optimal modularity allows us to identify four communities shown in Chart 1; each one has links of a particular number.

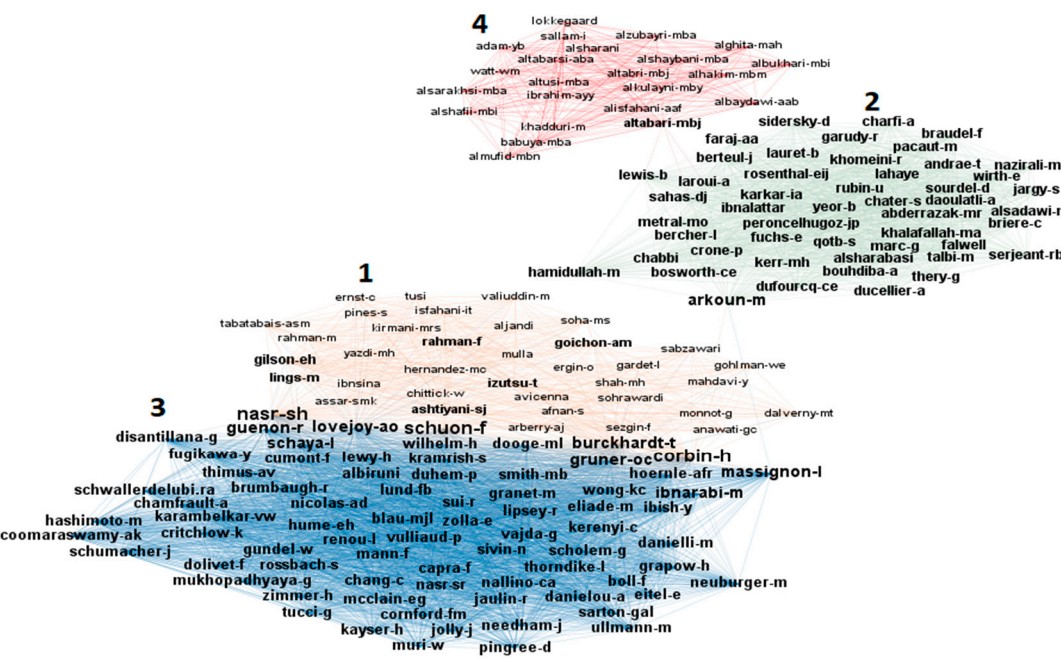

**Chart 1.** Asymmetric Co-citation Network of Articles Published Between 1980 and 1990.

A more eccentric community is marked with the number 4; this subgroup does not have a significant number of references. The authors cited in this subgroup are, for the most part, classics of Islamic theology. We are talking here about canonical works of Shi'a fiqh (Al-Shaykh Al-Mufid) or Sunni fiqh (al-Tabari). There are very few references to social sciences or contemporary orientalism in this community apart from a few exceptions (Montgomery 1974). This group of authors reflects strong isogamy. In addition to having a low co-citation level, this group shares very few connections with other communities.

Alongside this group, there is a second theoretical community marked with the number 2. Most of the references in this community are made up of contemporary authors from the Arab world (Laroui, Talbi). After reading these cited authors, one can note the

---

12　In this paper, the term 'orientalist' refers to the Western scholarly discipline that encompassed the study of the Asian and North African societies, during the 18th, 19th, and mid 20th century.

presence of some Islamist thinkers such as Sayyid Qutb or Ayatollah Khomeini. At the same time, one can also spot the presence of some classical reformers from the first half of the 20th century (Khalafallah, Abderrazak). The few citations outside the realm of Islamic references are from the social sciences (Braudel) or critical studies on Islam, which are strongly inspired by these disciplines (Bouhdiba). This community remains the one where the scholars of Islam are most present as it is marked by contemporary references from the Muslim world.

The two most numerically significant communities are those whose links are respectively marked with the numbers 1 and 3. With strong interconnections, these two groups refer to specific identities: the first one refers to oriental studies; however, the second is a much more heterogeneous group where philosophy and social science interact. Subgroup number 1 is primarily composed of references to philosophical works. The main authors cited are orientalists, whose studies deal with the Islamic mysticism of the Middle Ages (Izutsu, Lings, Ashtiyani). This community is supplemented by several metaphysicians (Gilson) or specialists in classical philosophy. For instance, one can cite Amélie Marie Goichon who is universally recognized as a specialist of Avicenna. The new generation of Islamic scholars are not very present in this group, but two exceptions can be highlighted: Mohamed Arkoun and Fazullah Rahman. This could be attributed to the fact that they hold institutional positions in France and the United States. The secondary references include a series of classical authors of Islamic philosophy (Avicenna, Nasir al-Din al-Tusi, Abu al-Faraj al-Isfahani) and some scholars who share an interest in mysticism (Anawati, Chittick, and Gardet). This subgroup is a good representation of the scientific dynamics within orientalism during the 1980s in that it shows interest in the canon of Islamic thought and a relative indifference towards the social sciences.

Community number 3 refers to a diverse collection of references. The authors who come closest to subgroup number 1 and who share the most connections with them are major scholars of Sufism (Guenon, Schuon, and Corbin Massignon). This first referential trend goes hand in hand with several references from social sciences and humanities (A.O. Lovejoy, Mann, Burckhardt, Eliade). In this subgroup, it is possible to find authors who understand religion in an alternative way (Ernest G. McClain). This community can be identified as the one that includes references outside the referential space of orientalism; that is to say, it refers to non-specialized authors.

From this statistical presentation, one can see some insightful trends within the scientific production of Muslim thinkers. There is a predominance of citations from the Western scientific world. The most dominant references reproduce the trends observed in classical Middle Eastern studies. Indeed, during the 1980s, the articles under study mostly reflect theoretical and methodological trends that are reproduced in Chart 1. There is a rather philological reading that focuses on the study of canonical texts, a historical and philosophical predisposition whose theoretical basis lies in phenomenology and metaphysics; finally, it is an approach that relies more on the social sciences but remains the approach of a minority. The only Islamic thinkers co-cited in communities number 1 and 3 refer to Sufi theology and philosophy, and mysticism.

In contrast, communities number 4 and 2 seem to be more focused on the normative and political issues of Islam. Group number 2 clearly shows concerns about the issue of Islamism. It regroups authors who want to reinforce their affiliations or who study political Islam. The group's links to community number 4 can be explained in light of this interest. In fact, in group number 2, we find authors who want to deconstruct or understand the references to Sharia (Islamic law) and to build authority in Islam. Therefore, there are co-citations with classic works on Fiqh and Sharia.

Two authors managed to position themselves between these two theoretical blocks: Muhammad Hamidullah and Mohamed Arkoun. These two scholars are both well recognized in orientalist circles. Their reputation allows them to operate in an international academic network. As polyglots, they share a strong bond with several Muslim countries. Likewise, their scientific output is also situated on the borders of these referential

spaces. Although exiled, they remain intimately engaged with Muslim societies and, more particularly, with its postcolonial future. It is through their lived experiences that these authors seek to rethink the relationship between politics and religion in Muslim countries. These commitments do not prevent them from being academic references when it comes to studying Islam.

It appears that there is a clear opposition between the two epistemological tendencies. The former is concerned with political issues and maintains a theoretical link wherever the Arab-Muslim referent (subgroup numbers 4 and 2) predominates. The latter trend refers to scientific research that is disinterested in any referent related to heterodox Islamic thought. Nevertheless, the last trend is the most significant in statistical terms as it seems to represent the predominance of orientalism in the reference space of our authors (the yellow and blue subgroups).

The influence of orientalism characterizes the first period (1980–1990). The latter is central to the references used by Islamic thinkers. This shows that this current of thought is not subject to systematic rejection but is reintegrated into a corpus where a small number of intellectuals from the Muslim world are associated. There is a notable lack of references from this geographical area. The few famous names echoing contemporary Islamic thought are intellectuals who occupy a special position in academic space (Mohamed Arkoun). Most of them are exiles who are forced to adapt to an academic world that remains foreign to them (Brisson 2008). At the same time, there were important references and participants bound by the theoretical space of their time who were incapable of establishing themselves as authorities. Their space of scientific possibility was structured by classical Islamic and Arabic studies. In all their works, they had to take their field's dominant approaches into account, even if it is from a critical perspective.

## 6. Second Period (1991–2001): Development of References in Humanities and Social Sciences (HSS)

For the second period, we analyzed a total of 80 articles published by the authors selected between 1991 and 2001. The Leuven algorithm identifies four groups marked with different numbers. From a disciplinary point of view, these communities seem more mixed during this period compared to the analysis of the literature of the 1980s. Indeed, the references selected in the four groups relate not only to Islam, but also to various subjects. Besides, authors from the orientalist corpus are less present.

A closer look at each of the groups mentioned above reveals some distinct scientific trends:

The community with the links marked with the number 2 brings together most of the scientific citations. We can find in this subgroup philosophers of science (Kuhn, Feyerabend), researchers working in different disciplines of pure sciences (Waldrop, Hawkins), and experts in the sociology of science and technology (Latour and Woolgar 1986).

It seems that this subgroup has a strong link with the natural sciences. As a result, it is not surprising to see Ziauddin Sardar holding a special status. As a multifaceted intellectual, Sardar studied physics and information sciences. As a journalist and a researcher specializing in scientific topics, he became known to the general public through numerous essays on Islam. This tendency to hold multiple positions on different issues would explain why this author sits astride several epistemic communities.

Alongside this first group, we identified another community whose links are marked on the graph with the number 1. This group mainly brings together a series of references to philosophy. One can also notice there a strong presence of Islamic thinkers. Thus, this group refers to a trend of interpreting Islam much more philosophically and, as a result, several political philosophers are members of this subgroup (Plato, Lefort, and Tocqueville). The reference to orientalism is much less present than in the previous time frame, as it is replaced by authors hailing from the social sciences (Inayatullah, Juergensmeyer).

Community number 3 is mostly composed of referents to the social sciences. The most common reference in this subgroup is the American anthropologist Clifford Geertz. He is the author of several books on religious practice in rural areas of Morocco and Indonesia.

This important citation reveals the main academic trend of this subgroup, namely the social sciences. Secondly, community number 3 seems to be where Islamic studies scholars (Izutsu, Pellat) are the most represented. They often come together with classic authors of Islamic thought (Ibn Qutayba).

On the border with group number 4, one finds two prominent figures of Islamic thought, Fazullah Rahman and Mohamed Arkoun. They seem to establish a link between the citations related to classical orientalism and a group concerned with normative issues in Islamic thought. Indeed, the citations of this subgroup refer to the topic of Sharia: first, we find several intellectuals who claim to be critical of this notion (Alazmeh, Hannafi). Then, there are several experts on the evolution of Islamic law (Hallaq, Asad). Ultimately, there are classical theologians of jurisprudence (Alashafii, Alashari). Community number 4 is supplemented by some general references to thinkers generally known as postmodern (Derrida, Baudrillard, Eliade).

Chart 2 shows great changes in the citation arrangements of our sample of authors. The collected referents seem to be more varied than in the first period under study (1980–1990). From a predominance of classical orientalism, we move to a corpus of citations in line with humanities and the social sciences. This new configuration of the referential space regarding the new generation of Islamic thinkers is informed, among others, by a series of factors that view Islam and Muslims as objects of scientific research. The decrease in classical Islamology's import on this issue is concomitant with the rise of Islam's influence in public debates. This situation generates a two-level movement of scientific decompartmentalization, first, in terms of the academic disciplines which re-appropriate Islam as an object of study in different scientific fields. Second, in terms of massification through the growing amount of research on this subject.

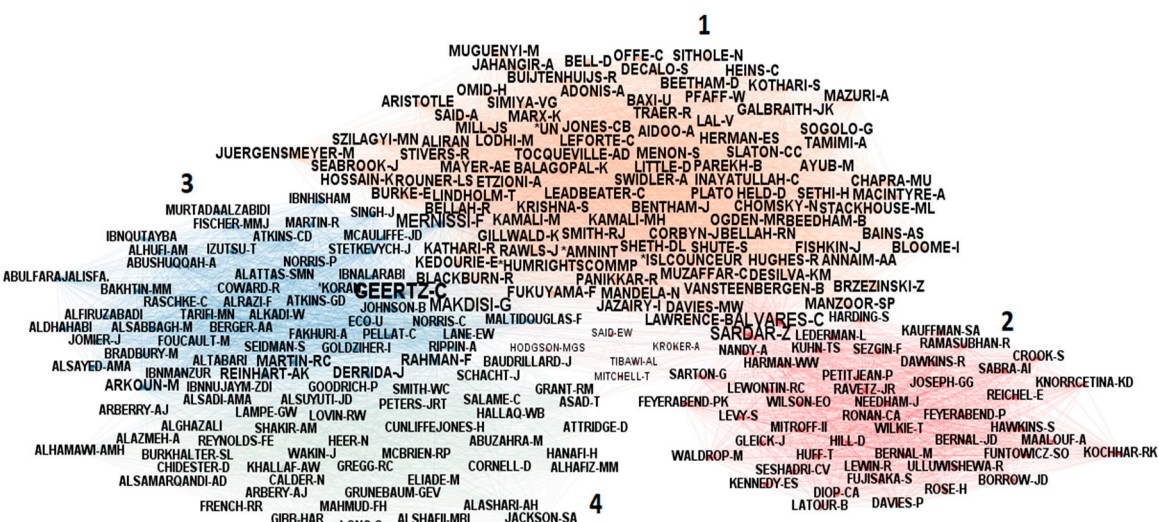

**Chart 2.** Asymmetric Co-citation Network of Articles Published between 1991 and 2001.

This general observation is reflected in the citations used by the new Islamic thinkers. For instance, we see that Clifford Geertz is the most prolific author in terms of co-citations. The American anthropologist is often brought forth as a critic of orientalism, or rather of its generalizing and essentializing tendencies. Through the study of specific areas, he puts into perspective the cultural permanence that has long been the subject of study for contemporary Islamology. This reference is indicative of a turning point where social sciences grew in influence. In addition to distancing themselves from orientalism, the other citations are multidisciplinary and therefore recognized in several fields. This new trend seems to reveal a desire to be acknowledged beyond Islamology. It shows the dynamics of academic affiliation as the authors under study seem to evolve towards alternative approaches to Islam.

A profound reworking of the theoretical space of Islamic thinkers characterizes the second period (1991–2001). There is a shift from a space dominated by the orientalist references to a space with a strong presence of social sciences and humanities. This drastic change shows a restructuring of the production of knowledge about Islam, where the question of its politicization becomes more and more central. Research centers are emerging in the Muslim world, often funded by ministries of foreign affairs or by philanthropic contributions, which leads to the predominance of political science over more traditional disciplines such as archeology (Roy 2001). This radical change in the intellectual field has had a strong impact on the intellectual output of Islamic thinkers. The intellectual debates of the time are mainly anchored in the idea of the end of history (Fukuyama 2006) in relation to the Muslim world. For instance, is Islamism perpetuating itself or are we dealing with an Islamic exception? This change is also perceived through a desire to study political Islam based on its ideological references. Hence, there is a major presence of ancient Islamic sources that were absent from the previous period.

## 7. Third Period (2002–2012): Proliferation of Sources and Mutual Recognition

The graphic layout of the third period comprises a total of 71 articles published by selected authors between 2002 and 2012. We retained all the nodes with a degree greater than 10. Thus, a total of 11 communities can be identified, and this is indicative of an increased plurality in citational dynamics compared to previous periods.

The three most representative communities are numbers 1, 3, and 8. By analyzing subgroup number 3, we can see the emergence of the general theme of political Islam. We first notice the presence of some key ideologues of Islamism (Qutb, Madudi, al-Banna). The subgroup also includes several intellectuals from the Muslim world who either inspired pan-Islamism (Nadwi, Benanbi) or criticized it (Ibrahim, Jada'an). This community is complemented by a series of references to Islamologists specializing in law or the political dimension of religion (Smith, Wilkinson).

Community number 8 is influenced by the issue of scientific discourse and its different "regimes of truth". This subgroup contains theoretical references centered on the analysis of contemporary transformations that societies have undertaken. For instance, it is worth mentioning the economist John Kenneth Galbraith and the sociologist Zygmunt Bauman. The rest of this community consists of a series of writers who take a critical look at the practice of science (Meadows, Ravetz, and Polimeni). It should also be noted that there is a strong presence of futures studies (Sardar, Polemini). Foresight is the process that aims, using a simultaneously rational and holistic approach, to get today ready for tomorrow. This multidisciplinary field of research aims to elaborate scenarios that are, in the researcher's perception, possible or impossible based on the analysis of available data (state of play, major trends, and phenomena of emergence). This type of study is much more present in Anglo-Saxon scholarship as it is strongly inspired by Ibn Khaldun's cyclical theory of "asabiyyah". Hence, the presence of this author is significant in community number 8.

Group number 1 deals with the debates concerning the place of Islam in the United States. It includes not only a set of scholars, but also engaged citizens who defend different perspectives on Islam. The debate over Islam has become crucial in American public life, notably when taking into consideration the attacks of 11 September. Several authors approach Islam from a national security perspective, including experts, journalists, and activists who adopt an anti-jihadist discourse (Geller, Emerson, and Ibrahim). In opposition to this standpoint, we note the presence of references to American researchers. This second group of references includes authors who want to highlight the plurality that Islam can take pride in (Kruzman, Bagby, Cainkar). There is also a critique of US security policy since the attacks of 11 September (Abdul-Ghafur, Greenberg). As such, most of this community originates in the United States and the citations mentioned address the question of Islam in that country.

In addition to these three groups, Chart 3 shows a series of secondary communities. First, there is subgroup number 2, which corresponds to French Islamology. One finds

there the classics of orientalism (Laoust, Massignion, Cahen) as well as political scientists and historians interested in Islam (Miquel, Etienne). This community is complemented by a cohort of North African researchers who are mostly interested in Islamism (Laghmani, Djait, Essid).

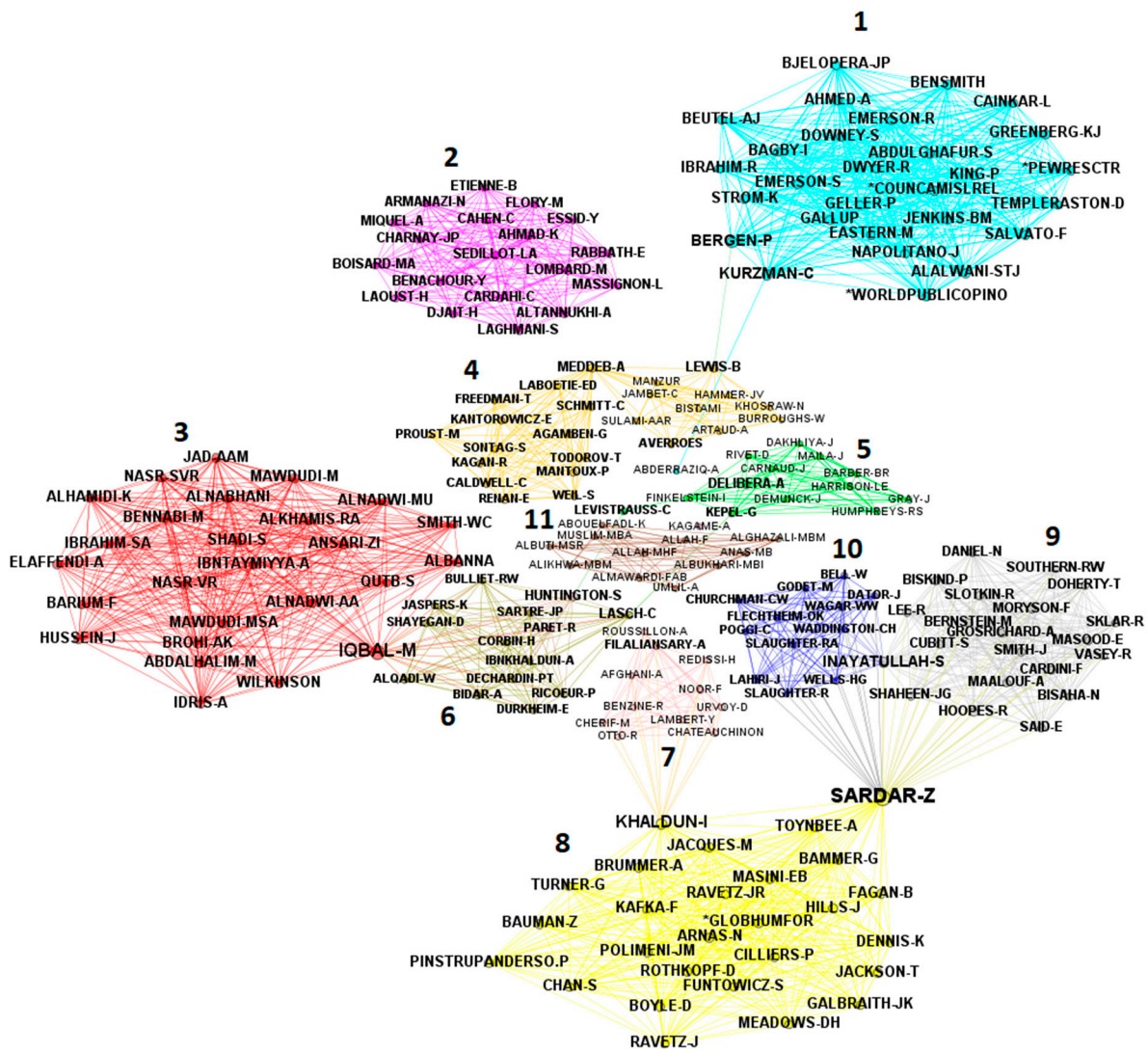

**Chart 3.** Asymmetric Co-citation Network of Articles published between 2002 and 2012.

Community number 4 includes two major types of references: those of political philosophy (Weil, Agamben, Schmit, and La Boetie) and those of literary theory (Proust, Caldwell, Todorov). This subgroup also includes a strong presence of ancient (Hammer-Purgstall) and contemporary (Lewis) Islamic references, as well as references to classical Islamic thought (Averroes, Bistami, Khosraw). The link between these two trends is established through the citation of an author from our initial corpus, Abdelwahab Meddeb, who is a writer, poet, and host of a Franco-Tunisian radio program. His work is inspired as much by orientalism as by literary criticism; his goal is to offer a reading of the Arab-Islamic cultural heritage that goes beyond religious fundamentalism, which he diagnoses as the "disease of Islam" (Meddeb 2013).

Most of community number 5 is based on Francophone social sciences. It includes some specialists of Islam (Rivet, Dakhlia, Kepel, Humphreys). There are also experts in international relations whose writings deal with Islam (Barber, Maila). The whole group

is complemented by a series of references to researchers addressing religious issues in an interdisciplinary way (De Libera, De Munck).

Community number 6 includes participants closely linked to French philosophy. Classic philosophical figures are found there, such as Sartre and Ricœur, alongside references to Islamologies that share these strong philosophical components (Corbin, Paret, and Jasper). There is also a group of intellectuals based in France who attempt to theorize about Islam, often using philosophical categories (Bidar, Shayegan).

Subgroup number 7 is that of twentieth-century Arab philosophy. It mainly includes specialists of this domain (Noor, Urvoy, Cherif) but also Islamologies that have produced reference works on Islamic modernism (Roussillion). Dr. Abdou Filaly Ansary is the author of a classic book on innovation and intellectual reform in connection with Islam. He notably participated in the introduction and translation of the works of classical and contemporary reformist writers.

Community number 9 is linked to cultural and American postcolonial studies. It contains references with a disciplinary focus on the study of Islam and the Arab world (Said 1979), but also essayists and researchers who study other topics (Slotkin, Biskind).

Subgroup number 10 is dominated by futures studies and its most cited author is Sohail Inayatullah, an influential scholar in this field, particularly through his research on Islam. This community is complemented by a series of references to futures studies either in the context of fiction or soft sciences and social sciences (Flechtheim, Churchman, Wells, and Waddington).

Subgroup number 11 comprises of citations relating to Islamic law. It includes classical authors of Islamic theology (al-Mawardi, al-Bukhari, al-Ghazali, Muslim) together with the contemporary researchers who study it (Abu el Fadl).

Chart 3 shows an increasing plurality of the type of citations used by our authors. The subject of Islam is addressed with greater disciplinary and linguistic diversity. It can be concluded that the interest in Islam is growing in the scientific field. Indeed, the citation reflects an academic space that allows for opportunities that would have been unlikely in previous periods. Thus, one can see futures studies being cited in great numbers, and an increasing number of references to authors who are not specialized in the study of Islam and Muslims. The most cited scholar of Islam in this period remains Ziauddin Sardar. His scientific versatility (skillfulness) allows him to be more and more present in the citational space. Moreover, he acts both as a leader in future studies and a preferred contributor when discussing issues affecting Muslims.

The third period (2002–2012) is the one in which Islamic thinkers are recognized. A fully-fledged community is established for a certain number of contemporary authors with multiple backgrounds and origins. These authors have a greater mutual acknowledgment as they cite each other more frequently than in previous periods. This new theoretical space owes a lot to the rise of Islamic studies as a discipline during the 2000s[13.] Following the 11 September attacks, research on Islam has seen an incredible boom in popularity (Kurzman and Ernst 2012; Mahmood 2006). This political crisis has led to shifts in the representations of Islam, which is no longer seen as an external phenomenon, but rather as a local issue in Western societies[14]. In this context, the field of Islamic studies has been granted massive endowments from Western governments and foreign funds[15]. The overall orientation of this discipline is to produce an alternative discourse to religious fundamentalism (Nanji 1997).

---

[13] For an overview of Islamic studies, see Azim Naji's book (1997).

[14] On the outcomes of crises on the intellectual field, see Frédérique Matonti's text on structuralism (Matonti 2005).

[15] Examples include Prince Alwaleed Bin Talal's Chair of Islamic Studies at Harvard University and Georgetown University's Chair for Muslim–Christian Understanding.

**8. Post-Islamism and Islamic Thought: The Birth of a New Scientific Discourse**

Post-Islamism is a concept that finds its roots in the works of Iranian American sociologist Asaf Bayat. It is defined as "an attempt to turn the underlying principles of Islamism on its head by emphasizing rights instead of duties, plurality in space of singular authoritative voice, historicity rather than fixed scripture, and the future instead of the past" (2007, p. 11). Specifically, it elucidates the major transformations undergone by Islamist movements as they abandon their projects of Islamicizing the state and imposing Sharia law. Moreover, the concept refers to a project that "emphasizes religiosity, individual choice, and human rights, as well as a plurality in place of a singular authoritative Islamist voice" (Bayat 2007, p. 10–11). In other words, it constitutes an attempt to reconcile Islam with liberal democracy, pluralism, and liberties. This observation is shared by several scholars. The new configurations of the Islamic intellectual space connected with the post-Islamist context have been the subject of many studies. These studies generally agree on attributing this change to the "upheavals rooted at the end of the Islamist utopia" (Zeghal 2008). This opposition to the ideologization of religion is a crucial element in the development of contemporary Islamic thought.

It is important to note that our research aspires to establish a new direction: the birth of output in the intellectual market. Post-Islamism, as a new condition, often leads to the development of new local ways of being, acting, and thinking in Europe and North America. The presence of a considerable Muslim population leads to a reformulation of religiosity (Cesari 2003; Haddad and Esposito 2000). The compromise with mainstream Western society can result in rethinking religious practice. Indeed, secularism has an impact on the religious practice of believers. European or North American Muslims are looking for intellectual authorities with whom they can identify. Therefore, in this new framework, Islamic thought is no longer premised on the deconstruction of religious "orthodoxy" as was previously the case. Rather, it is about understanding practices and interpretations that are anchored in the reality of pluralistic and individualistic societies[16]. Therefore, it is a context where the discourse of contemporary Islamic thinkers is adapted to the culture of this new audience.

The contribution of our research is to analyze the influence of this context on the scientific field. To do so, we have chosen to use bibliometrics. Citation is a good indicator of the nature of the references used by scholars. Co-citation networks, on the other hand, provide us with the epistemic universe of Islamic thinkers. This statistical analysis helps to elucidate the scientific evolution of the selected intellectuals in different historical contexts and in light of socio-political changes in the international arena.

The three-level periodization of the theoretical space of Islamic thinkers reveals many developments regarding important issues. In 1980, the primary references used by these Islamic thinkers were orientalist, which shows that their main concern remained academic. They engaged, sometimes critically, with oriental studies. We notice in the network the importance of reference to classical works of Islamic thought and orientalism. The 1990s witnessed a big change in the theoretical world of Islamic thinkers as social sciences— especially political science—became more hegemonic. Simultaneously, the relation between Islamism and democracy became a central theme. There is scientific literature that emerged during this period which dealt with the links or oppositions between liberalism and Islam (Kurzman 1998; Binder 1988). The third period (2002–2013) shows the presence of more links between the Islamic thinkers. We also note an increased plurality of references in terms of disciplines. This can be explained by the increase in intellectual and scientific interest following the 11 September attacks.

This empowerment of religious reference, which is a result of the post-Islamist period, could thus be understood through the dynamics of the academic world. The bibliometric analysis shows that the recognition of the authors studied in the scientific field is achieved simultaneously with the re-appropriation of a religious heritage. This interference that

---

16    A good example of this type of approach is Farid Essack's book on the Quran (Esack 1997).

combines different theoretical repertoires cannot merely be the consequence of individual wills. The scientific field, like any intellectual space, is characterized by struggles in which the positions of the actors are shaped both by the accumulated symbolic capital and the structure of the field they are invested in (Bourdieu 1971). It is by taking this competitive framework into account that we will be able to understand the recent reorientations of Islamic thought.

The three historical periods (the 1980s, 1990s, and 2000s) witnessed a transformation in the type of references used by the Islamic thinkers. The post-Islamist context enables contemporary Islamic thought to exist scientifically. The recent popularity enjoyed by the selected authors is achieved thanks to their desire to be placed in a new niche, which requires the building of religious credibility through making extensive references to the reform of Islam. This issue, therefore, seems to be mainly situational. In the 1980s, the references used referred to erudite knowledge, but gradually Islam became the subject of a broader debate. This evolution has clear repercussions on the theoretical universe of Islamic thinkers. From the 2000s onwards, references to public debate began to appear: for example, we cite the historian Alain De Libera an important actor in the Affair of Aristote au Mont Saint-Michel (Aristotle at Mont-Saint-Michel) in France[17], or the historian of the Middle East and expert on Islam, Bernard Lewis (Lewis 2002), who was part of the American debate on the intervention in Iraq (2003). The issues linked to Islam and Muslims are shifting from international questions to debates that directly affect the Euro-American space. This reconfiguration seems to favor the growth of Islamic studies and to produce an alternative discourse on Islam (Subject Centre for Languages, Linguistics and Area Studies and Subject Centre for Philosophical and Religious Studies 2008). At the same time, Islamic thinkers have to give their opinions on public issues. Thereby, this propels them to adopt more normative positions. Some critics evoke the birth of academic theology (Kramer 2001; Hughes 2015). However, these intellectuals try to elaborate a vision of Islam in conformity with political liberalism to disqualify essentialist interpretations of this religion in post 11 September Western public debate[18]. We must also question the existence of a real current or school of thought that would refer to a renewed Islamic thought. Indeed, the new demand produced by the field of Islamic studies reinforces a phenomenon of importation that contributes to recognizing these intellectuals as a religious movement. Thus, this may be a factor that could explain the increase in the number of monographs offering collective portraits of these thinkers[19]. This new academic exegesis enables the development of a reference that endorses the importation of religious frameworks into the scientific field, while relativizing essentialist discourses on Islam. Therefore, this sequence of appropriation could be regarded as a prelude to the formation of an established current of contemporary Islamic thought; a proposition, due to the limitations of our research, we can neither confirm nor refute.

**Author Contributions:** M.A.B.: conceptualization, methodology, data curation, formal analysis, writing, validation, original draft preparation. H.B.L.: writing, review and editing. Both authors have read and agreed to the published version of the manuscript.

**Funding:** This research received no external funding.

**Institutional Review Board Statement:** Not applicable.

**Informed Consent Statement:** Not applicable.

**Data Availability Statement:** Not applicable.

**Conflicts of Interest:** The authors declare no conflict of interest.

---

17  This controversy, opposing a group of 56 academics and other researchers in a polemic on the contribution of the Arab-Muslim world to the Western world, was triggered by the publication of Sylvain Gouguenheim Aristote au Mont Saint Michel (Gouguenheim 2008).

18  For a broader view of the debate between religious study and Islamic theology see (Larsson 2018).

19  This type of symbolic re-appropriation can be observed in the monographs, but also in the preface of the "Readers" about contemporary Islamic thought. It deserves further study.

## Appendix A.

**Table A1.** List of the Articles Selected for Bibliometrics (All the Titles of the Articles and Journals are Translated to English).

| Author | Year | Title (In English) | Journal |
|---|---|---|---|
| AbuZayd-N | 2010 | The "Others" in the Qur'an: An Hermeneutical Approach | Philosophy & Social Criticism |
| Akhtar-S | 1988 | Is There an Epistemic Parity Between Faith and Rejection? | Southern Journal of Philosophy |
| Akhtar-S | 1991 | Sadomasochism in the Perversions | Journal of the American Psychoanalytic Association |
| AnNaim-A | 1994 | What Do We Mean by Universal? + The Concept of Human-Rights in the Islamic World | Index on Censorship |
| AnNaim-A | 2000 | Human Rights and Islamic Identity in France and Uzbekistan: Mediation of the Local and Global | Human Rights Quarterly |
| Arkoun-M | 1980 | Reading Surah-18 | Annales-economies societes civilizations |
| Arkoun-M | 1981 | Brief Reflections on the Jihad Theme | Etudes theologiques et religieuses |
| Arkoun-M | 1989 | Explorations and Responses—New Perspectives for a Jewish–Christian–Muslim Dialogue | Journal of Ecumenical Studies |
| Arkoun-M | 2007 | The Answers of Applied Islamology | Theory, Culture & Society |
| Arkoun-M | 1984 | Positivism and Tradition in an Islamic Perspective–Kemalism | Diogenes |
| Arkoun-M | 1986 | Islamic Discourse, Orientalist Discourse, and Scientific Thought | Comparative Civilizations Review |
| Arkoun-M | 1986 | Two Mediators of Medieval Thought + Averroes and Maimonides | UNESCO courier |
| Arkoun-M | 2005 | Thinking in the Mediterranean Arena Today | Diogenes |
| Arkoun-M | 1987 | Reflections on the Concept of Islamic Reasoning | Archives de sciences sociales des religions |
| Arkoun-M | 1987 | The Unity of Man in Islamic Thought | Diogenes |
| Arkoun-M | 1998 | From Inter-Religious Dialogue to the Recognition of the Religious Phenomenon | Diogenes |
| Bidar-A | 2006 | Caricatures of Mahomet | Esprit |
| Bidar-A | 2003 | Letter of a European Muslim. Europe and the renaissance of Islam | Esprit |
| Bidar-A | 2010 | The "Outsiders of Islam" | Diogenes |
| Bidar-A | 2007 | Our Responsibilities for Islam | Esprit |
| Bidar-A | 2011 | Mohammed Arkoun and the Question of the Foundations of Islam | Esprit |
| Bidar-A | 2004 | The destiny of spiritual Europe | Esprit |
| Bidar-A | 2009 | Islam, Modernity, and the Future of Man | Esprit |
| Charfi-A | 2010 | Islam: The Test of Globalization | Philosophy & Social Criticism |
| Chebel-M | 2005 | In the Beginning There Was Mohammed | Historia |
| Chebel-M | 1999 | Saladin, The Moslem Saint Louis | Historia |
| Chebel-M | 2008 | Salvation by Abolition of Slavery | Historia |
| Chebel-M | 1998 | Circumcision Before the Time of Abraham | Historia |
| Chebel-M | 1999 | Some Rites of Knighthood Borrowed from the Moslems | Historia |
| Chebel-M | 2005 | Mohammed | Historia |
| Chebel-M | 2005 | Research on Patriarchal Desperation | Historia |
| Chebel-M | 1998 | Circumcision: Three Explanations of the Rite | Historia |

**Table A1.** *Cont.*

| Author | Year | Title (In English) | Journal |
|---|---|---|---|
| Chebel-M | 2003 | "It is Necessary to Separate the Palace from the Mosque?": Twentieth-century Islam and Secularism | Historia |
| Chebel-M | 2012 | 150 Signs of the Quran | Historia |
| Chebel-M | 1998 | Women, Jews, Christians . . . What the Quran Really Says About Them | Historia |
| Diagne-SB | 2010 | 0 and 1 in Our Digital Age | Anthropological theory |
| Diagne-SB | 2004 | Islam and Philosophy: Lessons From an Encounter | Diogenes |
| Diagne-SB | 2011 | From the Tower of Babel to the Ladder of Jacob: Claude Imbert Reading Merleau-Ponty | Paragraph |
| Diagne-SB | 2011 | African Philosophy Must Define Itself in Terms of Intellectual Projects | Critique |
| Diagne-SB | 2010 | De fato mahometano: Leibniz and Muhammad Iqbal on Islamic Fatalism | Diogenes |
| Diagne-SB | 2011 | African Philosophy and African Charter of Human Rights | Critique |
| Diagne-SB | 1985 | On the Literary Character of Oral Literature—Reading of 'contes wolof du baol' | Komparatistische hefte |
| Diagne-SB | 2006 | The Life Force and the Utopia of the Post-Human | Diogenes |
| Diagne-SB | 2010 | In Praise of the Post-Racial Negritude Beyond Negritude | Third text |
| Diagne-SB | 2002 | Keeping Africanity Open | Public culture |
| ElFadl-KA | 1998 | Muslims and Accessible Jurisprudence in Liberal Democracies: A Response to Edward B. Foley's Jurisprudence and Theology | Fordham law review |
| ElFadl-KA | 2000 | Fox Hunting, Pheasant Shooting, and Comparative Law | American journal of comparative law |
| Engineer-AA | 1992 | Genesis of Communal Violence | Economic and political weekly |
| Engineer-AA | 1994 | Communal Violence and the Role of Police | Economic and political weekly |
| Engineer-AA | 1996 | Remaking of Muslim Identity in India: Fact or Myth? | Indian journal of social work |
| Engineer-AA | 1998 | Communal Violence, 1998—Shifting Patterns | Economic and political weekly |
| Engineer-AA | 1987 | Bohra Reformists and Galiakot Struggle | Economic and political weekly |
| Engineer-AA | 1988 | Gian-Prakash Committee Report on the Meerut Riots | Economic and political weekly |
| Engineer-AA | 1995 | On Bombay | Economic and political weekly |
| Engineer-AA | 2007 | Religion and Poverty: A Qur'anic Approach | Journal of dharma |
| Engineer-AA | 2009 | A Liberative Approach to Issues of Muslim Women in India | Journal of dharma |
| Engineer-AA | 1995 | Kashmir—Autonomy Only Solution | Economic and political weekly |
| Engineer-AA | 1987 | Ethnic Conflict in South-Asia | Economic and political weekly |
| Engineer-AA | 1993 | Bombay Riots—the 2nd Phase | Economic and political weekly |
| Engineer-AA | 1993 | The Bastion of Communal Amity Crumbles | Economic and political weekly |
| Engineer-AA | 1986 | Gujarat—Communal Violence and Police Terror | Economic and political weekly |
| Engineer-AA | 1988 | Marriage and Communalism | Economic and political weekly |
| Engineer-AA | 1989 | Communal Riots in Muzaffar-Nagar, Khatauli and Aligarh | Economic and political weekly |
| Engineer-AA | 1994 | Bangalore Violence—Linguistic or Communal? | Economic and political weekly |
| Engineer-AA | 1988 | The Lessons of Murshidabad | Economic and political weekly |

**Table A1.** *Cont.*

| Author | Year | Title (In English) | Journal |
| --- | --- | --- | --- |
| Engineer-AA | 1995 | Communalism and Communal violence 1994 | Economic and political weekly |
| Engineer-AA | 1995 | Bhagalpur Riot Inquiry Commission Report | Economic and political weekly |
| Engineer-AA | 1984 | Bombay-Bhiwandi Riots in a National Political Perspective | Economic and political weekly |
| Engineer-AA | 1984 | Philippines—the Struggle for a Separate Islamic State | Economic and political weekly |
| Engineer-AA | 1987 | Old Delhi in the Grip of Communal Frenzy | Economic and political weekly |
| Engineer-AA | 1988 | Capitalist Development and Ethnic Tension | Economic and political weekly |
| Engineer-AA | 1988 | Azad, Maulana and the Freedom Struggle | Economic and political weekly |
| Engineer-AA | 1990 | Muslims in a Multi-Religious Society | Economic and Political Weekly |
| Engineer-AA | 1991 | Press on Ayodhya Kar Seva | Economic and Political Weekly |
| Engineer-AA | 2004 | Islam, Women and Gender Justice (Shari'ah law) | Journal of Dharma |
| Engineer-AA | 1981 | Bibarsharif Carnage—A Field Report | Economic and Political Weekly |
| Engineer-AA | 1981 | Gujarat 1. Communal Riots in Godhra—A Report | Economic and Political Weekly |
| Engineer-AA | 1982 | Maharashtra—Behind the Communal Fury | Economic and Political Weekly |
| Engineer-AA | 1994 | Asangaon Riots—Not a Communal Disturbance | Economic and Political Weekly |
| Engineer-AA | 1981 | Revolution Going Awry | Economic and Political Weekly |
| Engineer-AA | 1981 | Minorities—Trouble at Aligarh Muslim University—A Report | Economic and Political Weekly |
| Engineer-AA | 1988 | Religion and Liberation | Economic and Political Weekly |
| Engineer-AA | 1989 | Communal Frenzy at Indore | Economic and Political Weekly |
| Engineer-AA | 1989 | Anti-Rushdie Disturbances in Bombay | Economic and Political Weekly |
| Engineer-AA | 1995 | Ligarh Riots—Unplanned Outburst | Economic and Political Weekly |
| Engineer-AA | 1997 | Communalism and Communal Violence, 1996 | Economic and Political Weekly |
| Engineer-AA | 1986 | Communal Holocaust in Amravti | Economic and Political Weekly |
| Engineer-AA | 1988 | Aurangabad Riots—Part of Shiv-Senas Political Strategy | Economic and Political Weekly |
| Engineer-AA | 1988 | Sectarian Clashes in Bombay | Economic and Political Weekly |
| Engineer-AA | 1988 | Theological Creativity of Azad, Abul, Kalam | Indian Literature |
| Engineer-AA | 1994 | The Islamic Outlook of Inter-Religious Dialogue | Journal of Dharma |
| Engineer-AA | 2004 | The Future of Secularism in India | Futures |
| Engineer-AA | 1982 | Baroda Riots—Wages of Political Corruption | Economic and Political Weekly |
| Engineer-AA | 1991 | Remaking Indian Muslim identity | Economic and Political Weekly |
| Engineer-AA | 1994 | Human Rights and Dawoodi Bohras | Economic and Political Weekly |
| Engineer-AA | 1995 | Communalism and Communal Violence in 1995 | Economic and Political Weekly |
| Engineer-AA | 1992 | Communal Conflict After 1950—A Perspective | Economic and Political Weekly |
| Engineer-AA | 1997 | Communal Violence in Maharashtra | Economic and Political Weekly |
| Engineer-AA | 1983 | From Nationalism to Communalism—Transformation of Malegaon | Economic and Political Weekly |

**Table A1.** *Cont.*

| Author | Year | Title (In English) | Journal |
|---|---|---|---|
| Engineer-AA | 1985 | Ahmedabad—From Caste to Communal Violence | Economic and Political Weekly |
| Engineer-AA | 1985 | Communal Fire Engulfs Ahmedabad Once Again | Economic and Political Weekly |
| Engineer-AA | 1989 | Communal Propaganda in Elections—A Landmark Judgment | Economic and Political Weekly |
| Engineer-AA | 1992 | Communal Riots in Ahmedabad | Economic and Political Weekly |
| Engineer-AA | 1992 | Benaras Rocked by Communal Violence | Economic and Political Weekly |
| Engineer-AA | 1984 | Understanding Communalism—Report on a Seminar | Economic and Political Weekly |
| Engineer-AA | 1986 | Maharashtra—Engulfed in Communal Fire | Economic and Political Weekly |
| Engineer-AA | 1991 | Communal Riots Before, During, and After Lok Sabha Elections | Economic and Political Weekly |
| Engineer-AA | 1987 | Contemporary Trends of Religious Commitment in Islam | Journal of Dharma |
| Engineer-AA | 1990 | Grim Tragedy of Bhagalpur Riots—Role of Police-Criminal Nexus | Economic and Political Weekly |
| Engineer-AA | 1991 | The Bloody Trail—Janmabhoomi, Ram and Communal Violence in Up | Economic and Political Weekly |
| Engineer-AA | 1992 | Sitamarhi on Fire | Economic and Political Weekly |
| Engineer-AA | 1996 | How Muslims Voted | Economic and Political Weekly |
| Esack-F | 1993 | Quranic Hermeneutics—Problems and Prospects | Muslim World |
| Esack-F | 1988 | Three Islamic Strands in the South-African Struggle for Justice | Third World Quarterly |
| Hanafi-H | 1987 | Life in Peace—An Islamic Perspective | Bulletin of Peace Proposals |
| Hanafi-H | 2006 | Whose Order? Whose Millennium? Notes on Sorensen | Cooperation and Conflict |
| Meddeb-A | 2003 | What Should One Expect from a War? (An Islamic Writer Reflects on Iraq) | Esprit |
| Meddeb-A | 2011 | Islam in Tristes, Tropiques, Ramblings and Lucidity | Esprit |
| Meddeb-A | 1991 | Reflections on the Gulf War and Arab Theology | Esprit |
| Meddeb-A | 1995 | A Tunisian Writer Reflects on the Loss of Historical Identity and Islamic Fundamentalism in Present-Day Algeria | Esprit |
| Meddeb-A | 2002 | Elite and Populace in the Islamic World: The Ideology of Political Terror | Esprit |
| Meddeb-A | 2006 | Counter-Preaches | Esprit |
| Meddeb-A | 2002 | A Primer in Saudi Arabia's Wahhabite Form of Islam | Du-die zeitschrift der kultur |
| Meddeb-A | 2004 | Ways of Contraband | Esprit |

**Table A1.** *Cont.*

| Author | Year | Title (In English) | Journal |
|---|---|---|---|
| Meddeb-A | 1998 | Kateb Yacine: Abd el-Kader and the Algerian Independence | Europe-revue litteraire mensuelle |
| Meddeb-A | 1999 | Wanderer and Polygraphist (Arabic Culture, Poetics) | Boundary 2—An International Journal of Literature and Culture |
| Meddeb-A | 2006 | About the Disease of Double Genealogy | Expressions Maghrebines |
| Meddeb-A | 2012 | The Future of Freedom in Tunisia Bourguiba, Under the Guise of Inventory | Esprit |
| Meddeb-A | 2013 | The Sublime in "le fou d'elsa" | Coloquio-letras |
| Meddeb-A | 1996 | Art and Trance—Cultural Contribution and Exchange in an Increasingly International Art World | Esprit |
| Meddeb-A | 1994 | A Non-European Muslim Examines the Limits of European Art | Esprit |
| Meddeb-A | 2004 | Europe: The Conditions of the Universal (Contemporary Political Situation) | Esprit |
| Muzaffar-C | 2011 | The Long Journey to the Just: My Life, My Struggle | Inter-Asia Cultural Studies |
| Muzaffar-C | 1988 | Aliran + Malaysian Monthly Journal | Index on Censorship |
| Muzaffar-C | 1995 | From Human Rights to Human Dignity | Bulletin of Concerned Asian Scholars |
| Muzaffar-C | 2005 | Declaration of Jury of Conscience World Tribunal on Iraq—Istanbul 23–27 June 2005 | Feminist Review |
| Muzaffar-C | 2005 | The Relationship Between Southeast Asia and the United States: A Contemporary Analysis | Social Research |
| Nasr-SH | 2004 | Ours is Not a Dead Universe | Parabola—Myth, Tradition and the Search for Meaning |
| Nasr-SH | 2008 | The Sacred Foundations of Justice in Islam | Parabola—Myth, Tradition and the Search for Meaning |
| Nasr-SH | 1992 | The World-View and Philosophical-Perspective of Hakim Nizami-Ganjawi + Muslim Gnosis in the Persian Poetry of Nizami | Muslim World |
| Nasr-SH | 1982 | The Spiritual Significance of Jihad | Parabola—Myth, Tradition and the Search for Meaning |
| Nasr-SH | 2007 | The Mystery of the Earth | Parabola—Myth, Tradition and the Search for Meaning |
| Nasr-SH | 1994 | The One in Many + Religious Pluralism | Parabola—Myth, Tradition and the Search for Meaning |
| Nasr-SH | 1983 | Reflections on Islam and Modern Thought | Studies in Comparative Religion |
| Nasr-SH | 1980 | Ibn Sina Prophetic Philosophy | Cultures |
| Nasr-SH | 2000 | The Sufi Master: The Spiritual Testament of Shams al-Urafa (The "Sun of the Gnostics") | Parabola—Myth, Tradition and the Search for Meaning |
| Nasr-SH | 1998 | Islamic–Christian Dialogue—Problems and Obstacles to be Pondered and Overcome | Muslim World |
| Nasr-SH | 1980 | The Male and Female in the Islamic Perspective | Studies in Comparative religion |
| Nasr-SH | 1984 | With Burckhardt, Titus at the Tomb of Ibn-Arabi | Studies in Comparative religion |
| Nasr-SH | 1987 | Response to Kung, Hans Paper on Christian–Muslim Dialogue | Muslim World |
| Nasr-SH | 1989 | Existence (Wujud) and Quiddity (Mahiyyah) in Islamic Philosophy | International Philosophical Quarterly |
| Nasr-SH | 1981 | Progress and Evolution—A Reappraisal from the Traditional Perspective | Parabola—Myth, Tradition and the Search for Meaning |
| Nasr-SH | 1984 | The Role of the Traditional Sciences in the Encounter of Religion and Science: An Oriental Perspective | Religious Studies |
| Nasr-SH | 1985 | Response to Dean Thomas Review of "Knowledge and the Sacred" | Philosophy East & West |
| Nasr-SH | 1999 | Toward the Heart of Things (Excerpted from, The "Encounter of Man and Nature. The Spiritual Crisis of Modern Man") | Parabola—Myth, Tradition and the Search for Meaning |
| Nasr-SH | 2002 | Justice to the Enemy: The Heart of Islam—Not All Wars are Jihads | Parabola—Myth, Tradition and the Search for Meaning |

**Table A1.** *Cont.*

| Author | Year | Title (In English) | Journal |
|---|---|---|---|
| Osman-F | 1998 | Monotheists and the "Other": An Islamic Perspective in an Era of Religious Pluralism | Muslim World |
| Osman-F | 2003 | Mawdudi's Contribution to the Development of Modern Islamic Thinking in the Arabic-Speaking World | Muslim World |
| Ramadan-T | 2005 | At Home in Europe. Participation instead of Separation: European Muslims Must Learn to Live in the West Without Denying Their Beliefs | Internationale politik |
| Ramadan-T | 1996 | Islam and Modernity in Conflict | Etudes theologiques et religieuses |
| Ramadan-T | 2013 | European Muslims: Facts and Challenges | European Political Science |
| Ramadan-T | 2008 | Reading the Koran (Book of All Muslims) | New York Times Book Review |
| Ramadan-T | 2013 | The Challenges and Future of Applied Islamic Ethics Discourse: A Radical Reform? | Theoretical Medicine and Bioethics |
| Sachedina-A | 1980 | Al-Khums—The 5th in the Imami Shii Legal System | Journal of Near Eastern Studies |
| Sachedina-A | 2010 | Prudential Concealment in Shi'ite Islam a Strategy of Survival or a Principle? | Common Knowledge |
| Sachedina-A | 1980 | Al-Khums—The 5th in the Imami Shii Legal System | Journal of Near Eastern Studies |
| Sachedina-A | 2010 | Prudential Concealment in Shi'ite Islam a Strategy of Survival or a Principle? | Common Knowledge |
| Sachedina-A | 2000 | Guidance or Governance? A Muslim Conception of "Two-Cities" | George Washington Law Review |
| Safi-LM | 2011 | Religious Freedom and Interreligious Relations in Islam: Reflections on Da'wah and Qur'anic Ethics | Review of Faith & International Affairs |
| Sardar-Z | 2010 | Freeze Framing Muslims: Hollywood and the Slideshow of Western Imagination | Interventions—International Journal of Postcolonial Studies |
| Sardar-Z | 1995 | Can Small Countries Survive the Future? | Futures |
| Sardar-Z | 1992 | The Future of Eastern Europe—Lessons from the Third-World | Futures |
| Sardar-Z | 1993 | Do Not Adjust Your Mind—Postmodernism, Reality and the Other | Futures |
| Sardar-Z | 1981 | Between GIN and TWIN—Meeting the Information Needs of the Third-World | Aslib Proceedings |
| Sardar-Z | 1991 | Total Recall—Aliens, Others and Amnesia in Postmodernist Thought | Futures |
| Sardar-Z | 2002 | The "Saudi Sandwich"—Multilayered Saudi Society and Prejudices Against Foreigners | Du-die zeitschrift der kultur |
| Sardar-Z | 1992 | On Serpents, Inevitability and the South Asian Imagination | Futures |
| Sardar-Z | 1997 | Coming Home—Sex, Lies and all the 'I's in India | Futures |
| Sardar-Z | 2010 | Welcome to Postnormal Times | Futures |
| Sardar-Z | 1996 | Natural Born Futurist | Futures |
| Sardar-Z | 1994 | Complexity—Fad or Future | Futures |
| Sardar-Z | 1992 | Terminator-2—Modernity, Postmodernism and the Other | Futures |
| Sardar-Z | 1996 | The Future of Democracy and Human Rights—An Overview | Futures |
| Sardar-Z | 2010 | The Namesake: Futures; Futures Studies; Futurology; Futuristic; Foresight—What's in a Name? | Futures |
| Sardar-Z | 1993 | Colonizing the Future—The Other Dimension of Futures Studies | Futures |
| Sardar-Z | 1995 | Alt. Civilizations FAQ—Cyberspace as the Darker Side of the West | Futures |
| Sardar-Z | 1997 | A Futurist Beyond Futurists: The A, B, C, D (and E) of Ashis Nandy | Futures |

**Table A1.** *Cont.*

| Author | Year | Title (In English) | Journal |
|---|---|---|---|
| Sardar-Z | 2007 | Garden Snake (Reprinted from the San Francisco Chronicle, 16 December 2006) | Landscape Architecture |
| Sardar-Z | 1993 | Paper, Printing and Compact Disks—The Making and Unmaking of Islamic Culture | Media Culture & Society |
| Sardar-Z | 1994 | Conquests, Chaos and Complexity—The Other in Modern and Postmodern Science | Futures |
| Seddik-Y | 2010 | Mists and Turbulence in the 'Sunni' Ocean | Diogenes |
| Seddik-Y | 2013 | Single Machine Scheduling with Delivery Dates and Cumulative Payoffs | Journal of Scheduling |
| Seddik-Y | 1998 | Have We Ever Read the Koran? | Esprit |
| Talbi-M | 2000 | A Record of Failure | Journal of Democracy |
| Talbi-M | 1988 | Possibilities and Conditions for a Better Understanding Between Islam and the West | Journal of Ecumenical Studies |
| Talbi-M | 1999 | Saint Louis: See Tunis and Die (the French Monarch's Last Crusade) | Histoire |
| Talbi-M | 1982 | Louis, Crusader-Saint—A Tunisian Death | Histoire |
| Wadud-A | 2000 | Farm Household Efficiency in Bangladesh: A Comparison of Stochastic Frontier and Dea Methods | Applied Economics |
| Wadud-A | 2000 | Roundtable, Feminist Ideology and Religious Diversity—Feminist Theology, Religiously Diverse Neighborhood or Christian Ghetto? Part 4 | Journal of Feminist Studies in Religion |

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
