# Peer review of "Post-Islamism and Intellectual Production: A Bibliometric Analysis of the Evolution of Contemporary Islamic Thought"

_religions, doi:10.3390/rel12010049_

Round 1

Reviewer 1 Report

The article provides interesting data and a fresh and innovative take on the topic, but I do have some major concerns and this is mostly about the general frame and some terms that are used as if they are generally accepted terms in the academe. In addition, some descriptions seem to be too easy, viz. too simple, particularly when it comes to the evaluation of generic terms such as „fundamentalism“, „Islamism“ or „reformist“ thinkers/movement and their general agenda. The situation is much more complex.

The term „post-Islamism“ is introduced without any further explanation, but simply borrowed from Roy as a more or less fixed term. I doubt whether there is a general consensus regarding the definition of the term, not to speak of the term being accepted in the academe. There is need of clarification. In addition, in note 1 it is stated that Roy wouldn’t support the current thesis here. This is a kind of contradiction. I suggest to discuss the term post-Islamism in more detail and/or search for a more neutral one.

On p. 2 the term „fundamentalism“ is used as if it is also a generally accepted and easily applicable term. But „Post-Islamism“ (if we use the term at all) is not a mere counterprogram to „fundamentalism“. This is way to easy. This is also true for the approach to the emergence of „Islamism“ that is presented as a general backdraw on reformist thinkers. But the situation is much more complex and there was always a mix of different approaches. In addition, major aspects of the teachings of the early reformists of the 19th century were important for the development of „Islamism“ (if we use the term at all in this too general way).

The sample of authors is based on a rather strict approach and the result is – at least in my opinion  - a kind of „self-fulfilling prophecy“: it is mainly Western styled academics that become part of the sample and I cannot see any broader approach to intellectuals and academics in the Islamic world. Maybe the sample should be broadened.

Author Response

Thank you for your comments.

We have elaborated more on the concepts of Post-Islamism and fundamentalism. 

Reviewer 2 Report

”Post-Islamis and Intellectual Production: A Bibliometric Analysis of the Evolution of Contemporary Islamic Thought” is interesting and the article produces new data that is of relevance for the study of religions (Islam specifically). The text also produces a meta-analysis of a specific academic field by the help of a bibliometric analysis.

However, the author/s could benefit from making some of the methodological starting points clearer. For instance, the vocabulary used in the selection and classification process should be more explicit. What does “orientalism/orientalists”, “Muslim intellectuals” and even “Muslim authors” mean and how are they used and how are they understood in the process when selecting authors and text that should be included in the analysis? For instance, when talking about “Muslim authors” does that indicate that a person self-identify as a believer or something else? What does it mean that some authors “understand religion in an alternative way” (p. 9, line 289-290)? Why is only French and English texts included in the analysis, why not, for instance, German or Dutch? To put it differently, the use of languages in the selection process should be motivated and possible problems with a specific selection (and exclusion of other languages) should be addressed critically.  

The major problem with the article is, however, the use of color in the charts. When printing out the article all graphs and charts are black-and-white and this is a serious problem if you use color in the analysis. Maybe this is not a problem for the author/s, but for the publisher since the article becomes almost impossible to read if you have a black-and-white printout of the text. This problem most be solved before the article is proceeded to the next step.

I also think that the final analysis should be developed. For instance, it would be interesting to include a final discussion and maybe also a possible explanation for the changes that the author/s have found in the publication pattern/S. For instance, what about the impact of thinkers like Charles Taylor, Jürgen Habermas and Talal Asad on the concept of secularism and the idea that we live in a so-called post-secular age? It would also be interesting if the author/s discuss why the impact of post-Islamist thinkers diminished in the 21st century. What about the rise of Salafism and militant Islamism and even anti-Muslim sentiments? Have these developments pushed out the post-Islamists from the academic and public scene? If so, why? Even though the author/s does not agree with thinkers like, for instance, Aaron Hughes (see, for instance, the special issue of Culture and Religion - An Interdisciplinary Journal, Vol 18, No 1/2017 that deals with Aaron Hughes’ perspectives) or Martin Kramer (especially his Ivory Towers on Sand) these voices can also be seen as a response to some of the post-Islamist voices within the academia (i.e. Essack, Moosa, Safi, etcetera). The scholars who embraced a Post-Islamist perspective within the academia were often seen as a new form of theologians (i.e. scholars of Systematic Theology) and this sparked the old discussion/conflict between theology and religious studies at public funded university (see, for instance, Göran Larsson “Studying Islamic theology at European universities”, in Mohammed Hashas, Jan Jaap de Ruiter and Niels Valdemar Vinding (eds.), Imams in Western Europe. Developments, Transformations, and Institutional Challenges. Amsterdam: Amsterdam University Press, 2018, pp. 121-141 that address some of these discussions). The tension between "theology" and "Religious studies" should be addressed in the final analysis of the article, according to my opinion. By doing so the author/s could provide food for thought for further analysis and more studies.

Minor comments. On p. 14, footnotes 12, 13, 14 are put in the wrong font-size.

Besides these comments, the article is interesting and valuable, but I think that the author/s needs to revise the article for now. 

Author Response

Thank you for your comments.

The problem of the colors was fixed as we replaced them with numbers.

The conclusion was improved as well as concepts were further clarified. 

Round 2

Reviewer 1 Report

As far as I can see some of my concerns were addressed, particularly on the terminological level.

Author Response

Thanks a lot!

Reviewer 2 Report

The author/s have consider the comments and suggestions that I proposed.

Author Response

Thanks a lot
